# Sex differences in coronary artery calcium progression: The Korea Initiatives on Coronary Artery Calcification (KOICA) registry

Wonjae Lee[1,2], Yeonyee E. Yoon[1,2]*, Sang-Young Cho[3], In-Chang Hwang[1,2], Sun-Hwa Kim[2], Heesun Lee[1,4], Hyo Eun Park[1,4], Eun Ju Chun[1,5], Hyung-Kwan Kim[1,6], Su-Yeon Choi[1,4], Sung Hak Park[7], Hae-Won Han[8], Jidong Sung[9], Hae Ok Jung[10], Goo-Yeong Cho[1,2], Hyuk-Jae Chang[11,12]

1 Department of Internal Medicine, Seoul National University College of Medicine, Seoul, South Korea,
2 Department of Cardiology, Cardiovascular Center, Seoul National University Bundang Hospital, Seongnam, South Korea, 3 Division of Cardiology, Department of Internal Medicine, Gyeongsang National University Changwon Hospital, Changwon, South Korea, 4 Healthcare System Gangnam Center, Seoul National University Hospital, Seoul, South Korea, 5 Division of Radiology, Seoul National University Bundang Hospital, Seongnam, South Korea, 6 Department of Cardiology, Cardiovascular Center, Seoul National University, Seoul, South Korea, 7 Division of Radiology, Gangnam Heartscan Clinic, Seoul, South Korea, 8 Department of Internal Medicine, Gangnam Heartscan Clinic, Seoul, South Korea, 9 Division of Cardiology, Heart, Stroke & Vascular Institute, Samsung Medical Center, Seoul, South Korea, 10 Division of Cardiology, Department of Internal Medicine, College of Medicine, Seoul St. Mary's Hospital, The Catholic University of Korea, Seoul, South Korea, 11 Division of Cardiology, Yonsei Cardiovascular Center, Yonsei University Health System, Seoul, South Korea, 12 Division of Cardiology, Severance Cardiovascular Hospital, Yonsei University College of Medicine, Yonsei University Health System, Seodaemun-gu, South Korea

* yeonyeeyoon@gmail.com

**Data Availability Statement:** Due to ethical restrictions by the institutional review board committees of each participating healthcare center, the Korea Initiatives on Coronary Artery

## Abstract

Even with increasing awareness of sex-related differences in atherosclerotic cardiovascular disease (ASCVD), it remains unclear whether the progression of coronary atherosclerosis differs between women and men. We sought to compare coronary artery calcium (CAC) progression between women and men. From a retrospective, multicentre registry of consecutive asymptomatic individuals who underwent CAC scoring, we identified 9,675 men and 1,709 women with follow-up CAC scoring. At baseline, men were more likely to have a CAC score >0 than were women (47.8% vs. 28.6%). The probability of CAC progression at 5 years, defined as [$\sqrt{\text{CAC score (follow-up)}} - \sqrt{\text{CAC score (baseline)}}$] $\geq 2.5$, was 47.4% in men and 29.7% in women ($p<0.001$). When we stratified subjects according to the 10-year ASCVD risk (<5%, $\geq$5% and <7.5%, and $\geq$7.5%), a sex difference was observed in the low risk group (CAC progression at 5 years, 37.6% versus 17.9%; $p<0.001$). However, it became weaker as the 10-year ASCVD risk increased (64.2% versus 46.2%; $p<0.001$, and 74.8% versus 68.7%; $p = 0.090$). Multivariable analysis demonstrated that male sex was independently associated with CAC progression rate among the entire group ($p<0.001$). Subgroup analyses showed an independent association between male sex and CAC progression rate only in the low-risk group. The CAC progression rate is higher in men than in women. However, the difference between women and men diminishes as the 10-year ASCVD risk increases.

Calcification (KOICA) registry data underlying this study cannot be made publicly available, as public availability would compromise patient confidentiality and participant privacy. Therefore, access to aggregated data will be granted following review by the KOICA steering committee; data access requests can be sent to onlylhw1230@yuhs.ac.

**Funding:** The author(s) received no specific funding for this work.

**Competing interests:** The authors have declared that no competing interests exist.

## Introduction

Coronary artery disease (CAD) is the leading cause of death worldwide for both men and women [1]. Given the worldwide health and economic implications of atherosclerotic cardiovascular disease (ASCVD) in women, there is a strong rationale to sustain an effort to control major ASCVD risk factors and apply evidence-based therapies in women [2]. Currently, adverse trends in ASCVD risk factors among women are an ongoing concern. For example, in older adults, a higher percentage of women than men have hypertension, and the gap will likely increase in the aging society [3]. The prevalence of diabetes in women is also increasing, which exacerbates the overall risk of ASCVD [4]. Nevertheless, women are regarded as at lower risk for ASCVD than are men, and are not given aggressive preventive medications, such as statins, despite a similar benefit for both women and men [5–7].

Coronary artery calcium (CAC) is a characteristic of coronary atherosclerosis, and the detection and quantification of CAC significantly improves the risk stratification of ASCVD [8]. Especially, the importance of CAC imaging in women has been repeatedly proven, even in those with a low risk factor burden [9–12]. However, atherosclerosis is a dynamic process. While baseline CAC can be thought of as a single time point in atherosclerosis, the assessment of CAC progression provides insight into the dynamic atherosclerotic process in given individuals. Although male sex is a well-known risk factor for CAC progression [13], it has not been established whether CAC progression differs by sex. Therefore, we aimed to evaluate sex differences in CAC progression in a large cohort of asymptomatic individuals in the general population. In addition, we attempted to evaluate whether the sex difference varies according to the 10-year ASCVD risk.

## Methods

### Study population

This retrospective study was approved by Institutional Review Board of Seoul National University Bundang Hospital (and that of each participating institution) and conducted in accordance with the Declaration of Helsinki. The approval number was IRB No. B-1506/302-110. The need for informed consent from study participants was waived. The Korea Initiatives on Coronary Artery Calcification (KOICA) registry is an observational, retrospective, multicentre registry of Korean individuals who underwent CAC scoring as a part of a health check-up in a self-referral setting at six healthcare centres [14]. In the registry, 93,914 participants were enrolled between April 2003 and March 2017 (Fig 1). Among these, we identified 12,638 participants who underwent at least two CAC scans; however, 1,254 of these had incomplete data for the calculation of the 10-year ASCVD risk and were excluded. Finally, 11,384 participants remained for the analysis.

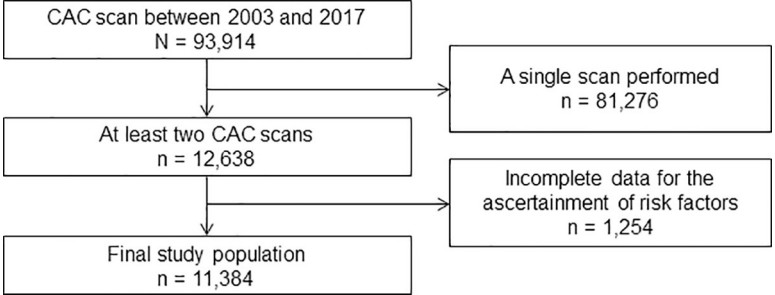

**Fig 1. Flow chart of the study participant selection process.** CAC, coronary artery calcium.

## Ascertainment of risk factors

During the health check-up, sociodemographic factors, risk profiles, and medication history were collected by a detailed questionnaire. All participants underwent clinical examinations, including a physical examination and laboratory tests. The predicted 10-year ASCVD risk was estimated by using the Korean Risk Prediction Model, which is a recalibration of the Pooled Cohort Equation specifically for the Korean Population [15, 16]. The 10-year ASCVD risk takes into account the participant's age, sex, ethnicity, total cholesterol, high-density lipoprotein (HDL) cholesterol, treatment for hypertension, systolic blood pressure, smoking, and the presence of diabetes. The study participants were classified into three groups according to their 10-year ASCVD risk: <5%, ≥5% and <7.5%, or ≥7.5% [17].

## Cardiac computed tomography (CT) acquisition and analysis

Multi-detector CT scanners used to assess CAC had at least 16 slices (Siemens 16-slice Sensation, Philips Brilliance 256 iCT, Philips Brilliance 40 channel multi-detector CT, and GE 64-slice Lightspeed). CAC scores were calculated using the Agatston method [18]. The square root transformed difference was calculated [√CAC score (follow-up)—√CAC score (baseline)], and CAC progression was defined as a square root transformed difference of >2.5 to minimize the effect of interscan variability [19, 20]. For participants with more than two CT scans, the square root transformed difference was calculated for each follow-up CT scan, and the earliest follow-up scan with demonstrated CAC progression was included in the analysis. The CAC progression rate was calculated as the annualized difference between the square root of the baseline and last follow-up CAC scores.

## Statistical analysis

Continuous variables are expressed as means ± standard deviation; categorical variables are expressed as proportions. The Student's t-test and Wilcoxon rank-sum test were used to evaluate group differences in normally distributed and non-normally distributed continuous variables, respectively. Categorical variables were compared using the $\chi 2$ test or Fisher's exact test, as appropriate. Kaplan-Meier curves were used to visualize and estimate the distribution of the time to CAC progression according to sex, with differences evaluated using the log-rank test. The earliest scan date of detected CAC progression was assigned as the occurrence of an event. We also evaluated Kaplan-Meier curves using a propensity matched cohort, with a matching ratio of 1:2 (women to men).

Univariable and multivariable linear regression analyses were performed to determine the effects of conventional coronary risk factors and the baseline CAC score on the annualized progression of the CAC score. The multivariable analysis was initiated with the following conventional risk factors (model 1): age, male sex, waist circumference, hypertension, dyslipidaemia, diabetes, current smoking, systolic blood pressure, low-density lipoprotein (LDL) cholesterol, HDL cholesterol, triglyceride, creatinine, high-sensitive C-reactive protein, and glycated haemoglobin (HbA1c). Body mass index, diastolic blood pressure, and estimated glomerular filtration rate were excluded from the multivariable analysis because of multicollinearity (variance inflation factor of 2). We also performed a multivariable analysis with the addition of the baseline CAC score to the conventional risk factors in model 1 (model 2). The results are expressed as regression coefficients (β) and the corresponding 95% confidence interval (CI). Stepwise regression under Akaike's Information Criterion was performed to determine the appropriate multivariate linear regression model. For subgroup analyses, the variables that remained in the final model of the multivariable analysis were included.

All reported p-values are 2-tailed, and $p \leq 0.050$ was considered statistically significant. All statistical analyses were performed using R Statistical Software/environment (version 3.4.3, The R foundation for Statistical Computing, Vienna, Austria).

## Results

### Baseline characteristics

The 11,384 study participants consisted of 9,675 men and 1,709 women. Baseline characteristics of the study participants are provided in Table 1. Although age was not significantly different between men and women, men showed a higher proportion of conventional cardiovascular risk factors, and a higher 10-year ASCVD risk, compared to that in women. Therefore, women tended to be classified into lower risk groups. While the percentages of women with 10-year ASCVD risks of <5%, ≥5% and <7.5%, and ≥7.5% were 74.7%, 11.4%, and 13.9%, those of men were 69.2%, 14.3%, and 16.6%, respectively ($p<0.001$). Women also showed lower baseline CAC scores compared to those in men.

**Table 1. Baseline characteristics.**

|  | All | Men | Women | P value |
|---|---|---|---|---|
|  | (n = 11,384) | (n = 9,675) | (n = 1,709) |  |
| Age | 51.4 ± 8.5 | 51.4 ± 8.2 | 51.7 ± 9.7 | 0.288 |
| Body mass index, kg/m$^2$ | 24.5 ± 2.7 | 24.8 ± 2.6 | 23.0 ± 3.0 | <0.001 |
| Waist circumference, cm | 86.2 ± 8.0 | 87.5 ± 7.1 | 77.8 ± 8.6 | <0.001 |
| Systolic blood pressure, mmHg | 119.5 ± 15.0 | 119.8 ± 14.8 | 117.5 ± 16.0 | <0.001 |
| Diastolic blood pressure, mmHg | 75.0 ± 10.5 | 75.8 ± 10.4 | 71.0 ± 10.7 | <0.001 |
| Hypertension, n (%) | 3534 (31.0%) | 3133 (32.4%) | 401 (23.5%) | <0.001 |
| Diabetes, n (%) | 1371 (12.0%) | 1246 (12.9%) | 125 (7.3%) | <0.001 |
| Dyslipidaemia, n (%) | 2483 (21.8%) | 2178 (22.5%) | 305 (17.8%) | <0.001 |
| Current smoking, n (%) | 3248 (28.5%) | 3121 (32.3%) | 127 (7.4%) | 0.072 |
| Haemoglobin, g/dL | 15.0 ± 1.3 | 15.3 ± 1.0 | 13.1 ± 1.1 | <0.001 |
| Total cholesterol, mg/dL | 197.3 ± 33.9 | 197.1 ± 33.6 | 198.4 ± 35.5 | 0.163 |
| HDL cholesterol, mg/dL | 53.4 ± 16.0 | 52.0 ± 15.2 | 61.6 ± 17.9 | <0.001 |
| LDL cholesterol, mg/dL | 121.9 ± 31.7 | 122.9 ± 31.3 | 116.1 ± 33.7 | <0.001 |
| Triglyceride, mg/dL | 142.2 ± 88.8 | 149.1 ± 91.0 | 103.3 ± 61.9 | <0.001 |
| Creatinine, mg/dL | 1.0 ± 0.2 | 1.0 ± 0.1 | 0.8 ± 0.1 | <0.001 |
| eGFR, ml/min/1.73 m$^2$ | 88.6 ± 13.8 | 88.3 ± 13.3 | 90.6 ± 15.9 | <0.001 |
| hs-CRP, mg/dL | 0.4 ± 1.8 | 0.4 ± 1.4 | 0.5 ± 3.1 | 0.089 |
| Fasting glucose, mg/dL | 97.7 ± 20.2 | 98.7 ± 20.7 | 92.1 ± 16.0 | <0.001 |
| HbA1C, % | 5.7 ± 0.7 | 5.7 ± 0.8 | 5.7 ± 0.6 | 0.427 |
| 10-year ASCVD risk, % | 4.6 ± 4.6 | 4.8 ± 4.6 | 3.9 ± 4.5 | <0.001 |
| <5% | 7,968 (70.0%) | 6,691 (69.2%) | 1,277 (74.7%) | <0.001 |
| ≥5%, <7.5% | 1,575 (13.8%) | 1,381 (14.3%) | 194 (11.4%) |  |
| 7.5%≥ | 1,841 (16.2%) | 1,603 (16.6%) | 238 (13.9%) |  |
| Baseline CAC score | 57.8 ± 187.6 | 62.7 ± 196.4 | 30.3 ± 123.3 | <0.001 |
| 0 | 6,271 (55.1%) | 5,050 (52.2%) | 1,221 (71.4%) | <0.001 |
| >0, <100 | 3,508 (30.8%) | 3,146 (32.5%) | 362 (21.2%) |  |
| ≥100 | 1,605 (14.1%) | 1,479 (15.3%) | 126 (7.4%) |  |

HDL: high density lipoprotein; LDL: low density lipoprotein; eGFR: estimated glomerular filtration rate; hs-CRP: high-sensitive C-reactive protein; CAC: coronary artery calcium; ASCVD: atherosclerotic cardiovascular disease; HbA1c, glycated haemoglobin

## Sex differences in CAC progression

The median duration between the initial and last follow-up scan was 2.8 years for men (interquartile range, 1.9–4.1 years) and 2.6 years for women (interquartile range, 1.9–4.1 years) (p = 0.464). During follow-up, 3,131 of 9,675 men (32.4%) and 264 of 1,709 women (15.4%) experienced CAC progression. The absolute increase in CAC score in men and women was 49.8±143.4 and 19.1±76.5 ($p<0.001$), respectively (Fig 2A). The mean CAC progression rate for men and women was 0.9±2.0 and 0.4±1.3 ($p<0.001$), respectively (Fig 2B). Comparison of the cumulative proportion of CAC progression between men and women revealed that the chance of CAC progression was higher in men than in women (proportion of CAC progression at 5 years, 47.4% vs. 29.7%; $p<0.001$) (Fig 3).

In subgroup analyses according to the 10-year ASCVD risk, absolute changes in the CAC score and annualized CAC progression rate increased as the 10-year ASCVD risk increased in both men and women (Fig 2). Interestingly, the difference between men and women decreased as the 10-year ASCVD risk increased. The cumulative proportion of CAC progression was also compared according to the 10-year ASCVD risk (Fig 4). In the subgroup with a 10-year ASCVD risk <5%, a difference in the CAC progression between women and men was evident (proportion of CAC progression at 5 years, 37.6% versus 17.9%, respectively; $p<0.001$) (Fig 4A). However, the difference was less prominent in the subgroup with a 10-year ASCVD risk ≥5% and <7.5% (proportion of CAC progression at 5 years, 64.2% versus 46.2%, respectively; $p<0.001$), and was further diminished in the subgroup with a 10-year ASCVD risk ≥7.5% (proportion of CAC progression at 5 years; 74.8% versus 68.7%, respectively; $p = 0.090$) (Fig 4B and 4C).

In the Kaplan-Meier curves from the propensity score matched cohort, men demonstrated a higher risk of CAC progression than did women (S1 Fig). In the subgroup analysis, men had a higher risk of CAC progression in the subgroups with a 10-year ASCVD risk <7.5% (S2A and S2B Fig), but the curves became similar between men and women in the subgroup with a 10-year ASCVD risk ≥7.5% (S2C Fig).

## Association of sex with CAC progression rate

Univariable linear regression analyses showed that almost all conventional risk factors, including male sex, were significantly associated with the CAC progression rate (Table 2). In the multivariable model with conventional risk factors (model 1), male sex remained as a significant

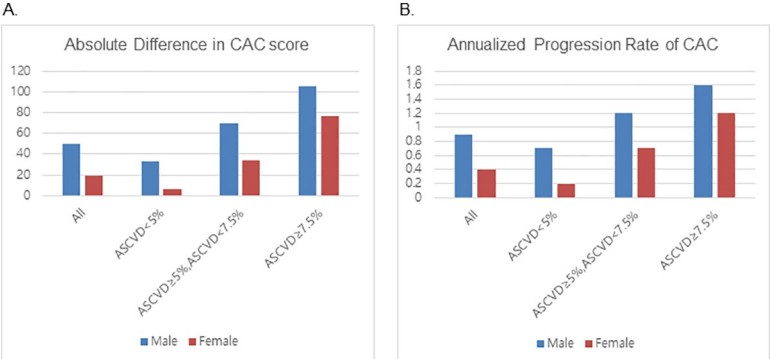

**Fig 2. CAC progression for men versus women according to the 10-year ASCVD risk score.** (A) Absolute difference in the CAC score between the first and last scanning sessions, (B) Annualized progression rate of CAC. CAC, coronary artery calcium; ASCVD, atherosclerotic cardiovascular disease.

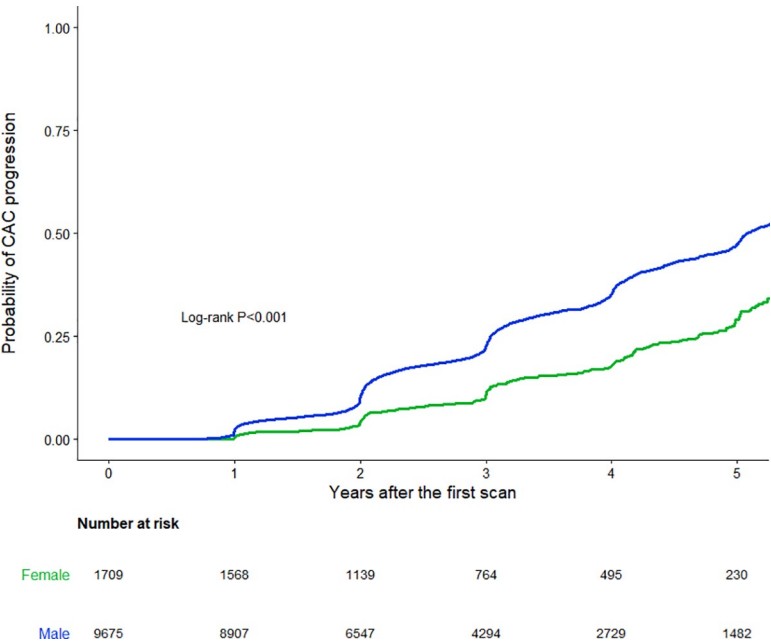

**Fig 3. Kaplan-Meier 5-year CAC progression for men versus women in asymptomatic Korean individuals.** CAC, coronary artery calcium.

predictor, along with age, waist circumference, hypertension, diabetes, hyperlipidaemia, current smoking, LDL cholesterol, HDL cholesterol, triglyceride, creatinine, and HbA1c (Table 3). Similar results were observed in the multivariable model with the additional inclusion of the baseline CAC score (model 2). In subgroup analyses according to the 10-year ASCVD risk, male sex was independently associated with the CAC progression rate in the subgroup with a 10-year ASCVD risk <5% (model 1: β = 0.397; 95% CI, 0.158 to 0.636; $p$ = 0.001 and model 2: β = 0.265; 95% CI, 0.030 to 0.500; $p$ = 0.028). However, male sex did not remain an independent predictor in the subgroups with a 10-year ASCVD risk ≥5% and <7.5% (model 1: β = 0.324; 95% CI, -0.190 to 0.838; $p$ = 0.217 and model 2: β = 0.264; 95% CI, -0.250 to 0.778; p = 0.313) and a 10-year ASCVD risk >7.5% (β = 0.220; 95% CI, -0.211 to 0.651; $p$ = 0.317 and model 2: β = 0.149; 95% CI, -0.272 to 0.570; $p$ = 0.494).

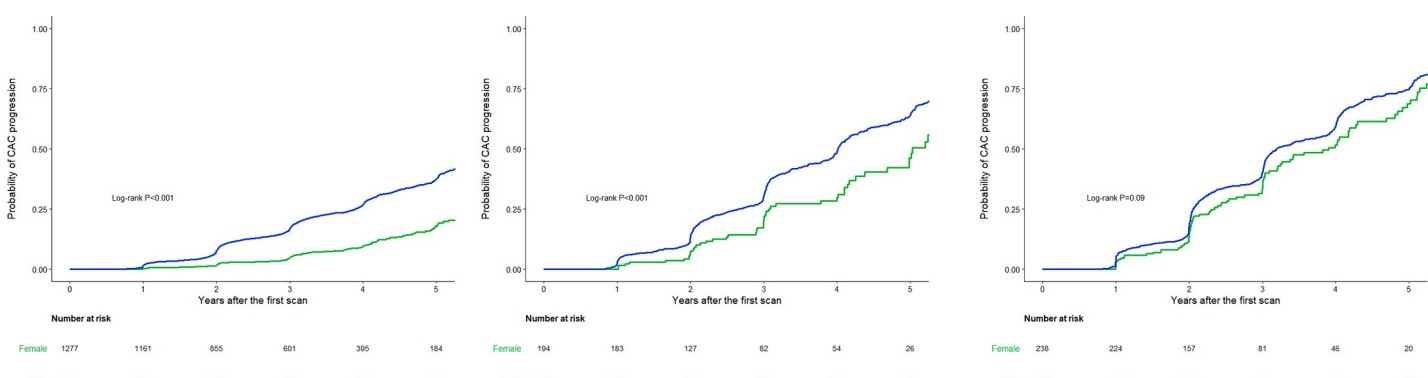

**Fig 4. Kaplan-Meier 5-year CAC progression for men versus women according to the 10-year ASCVD risk score.** (A) 10-year ASCVD risk <5%, (B) 5%≤ 10-year ASCVD risk <7.5%, (C) 10-year ASCVD risk ≥7.5%. CAC, coronary artery calcium; ASCVD, atherosclerotic cardiovascular disease.

**Table 2. Univariate analysis for factors associated with the CAC progression rate.**

| | Univariate | | | | |
| --- | --- | --- | --- | --- | --- |
| | **B** | **95% CI** | | | **P value** |
| Male sex | 0.576 | 0.051 | - | 0.576 | <0.001 |
| Age | 0.040 | 0.036 | - | 0.044 | <0.001 |
| Body mass index, kg/m$^2$ | 0.074 | 0.060 | - | 0.088 | <0.001 |
| Waist circumference, cm | 0.031 | 0.025 | - | 0.037 | <0.001 |
| Systolic blood pressure, mmHg | 0.007 | 0.005 | - | 0.009 | <0.001 |
| Diastolic blood pressure, mmHg | 0.007 | 0.003 | - | 0.011 | <0.001 |
| Hypertension | 0.667 | 0.589 | - | 0.745 | <0.001 |
| Diabetes | 0.846 | 0.736 | - | 0.956 | <0.001 |
| Dyslipidaemia | 0.555 | 0.469 | - | 0.641 | <0.001 |
| Current smoking | 0.162 | 0.084 | - | 0.240 | <0.001 |
| Haemoglobin, g/dL | 0.074 | 0.047 | - | 0.101 | <0.001 |
| Total cholesterol, mg/dL | 0.000 | -0.002 | - | 0.002 | 0.616 |
| HDL cholesterol, mg/dL | -0.005 | -0.007 | - | -0.003 | <0.001 |
| LDL cholesterol, mg/dL | 0.003 | 0.001 | - | 0.005 | <0.001 |
| Triglyceride, mg/dL | 0.002 | 0.002 | - | 0.002 | <0.001 |
| Creatinine, mg/dL | 0.79 | 0.578 | - | 1.002 | <0.001 |
| eGFR, ml/min/1.73 m$^2$ | -0.011 | -0.013 | - | -0.009 | <0.001 |
| hs-CRP, mg/dL | -0.033 | -0.055 | - | -0.011 | 0.003 |
| Fasting glucose, mg/dL | 0.011 | 0.009 | - | 0.013 | <0.001 |
| HbA1C, % | 0.301 | 0.246 | - | 0.356 | <0.001 |
| 10-year ASCVD risk | 7.670 | 6.908 | - | 8.432 | <0.001 |
| Baseline CAC score | 0.002 | 0.002 | - | 0.002 | <0.001 |

HDL: high density lipoprotein; LDL: low density lipoprotein; eGFR: estimated glomerular filtration rate; hs-CRP: high-sensitive C-reactive protein; CAC: coronary artery calcium; ASCVD: atherosclerotic cardiovascular disease; HbA1c, glycated haemoglobin

## Discussion

The present study demonstrated a sex difference in CAC progression in a large number of asymptomatic individuals. The probability of CAC progression was higher in men than in women, and the difference grew over time. However, in subgroup analyses according to the 10-year ASCVD risk, the sex difference diminished as the 10-year ASCVD risk increased. Additionally, although male sex was independently associated with CAC progression among the entire study cohort, it maintained an independent association with CAC progression only in the subgroup with a 10-year ASCVD risk <5.0%.

Data regarding differences in CAC progression between men and women are sparse. The results of the Multi-Ethnic Study of Atherosclerosis (MESA) study suggested more rapid CAC progression in men than in women, as CAC scores were higher in men than in women and increased more rapidly with age [21]. In addition, a previous study directly showed significantly greater progression of coronary atherosclerosis (defined as progression in the CAC score or coronary atherosclerotic plaque extent/burden) in men than in women [22]. However, when CAC progression was separately analysed, no sex difference was observed. This may be due to the fact that the study population in this previous report comprised patients with suspected CAD, and the women were significantly older than the men (57.5 versus 51.9 years, p<0.001). In contrast, the present study included asymptomatic men and women, and there was not a significant sex difference in age. Additionally, we evaluated the dynamic

**Table 3. Multivariate analysis for factors associated with the CAC progression rate.**

| | Model 1[a] | | | | Model 2[b] | | | |
|---|---|---|---|---|---|---|---|---|
| | β | 95% CI | | P value | β | 95% CI | | P value |
| Male sex | 0.396 | 0.186 | - 0.606 | <0.001 | 0.270 | 0.079 | - 0.459 | <0.001 |
| Age | 0.032 | 0.026 | - 0.038 | <0.001 | 0.020 | 0.014 | - 0.026 | <0.001 |
| Body mass index, kg/m$^2$ | | | | | | | | |
| Waist circumference, cm | 0.012 | 0.006 | - 0.018 | 0.002 | 0.010 | 0.004 | - 0.016 | 0.006 |
| Systolic blood pressure, mmHg | | | | | | | | |
| Diastolic blood pressure, mmHg | | | | | | | | |
| Hypertension | 0.321 | 0.211 | - 0.431 | <0.001 | 0.253 | 0.143 | - 0.363 | <0.001 |
| Diabetes | 0.452 | 0.266 | - 0.638 | <0.001 | 0.368 | 0.184 | - 0.552 | <0.001 |
| Dyslipidaemia | 0.256 | 0.138 | - 0.374 | <0.001 | 0.254 | 0.138 | - 0.370 | <0.001 |
| Current smoking | 0.239 | 0.123 | - 0.355 | <0.001 | 0.225 | 0.109 | - 0.341 | <0.001 |
| Haemoglobin, g/dL | | | | | | | | |
| Total cholesterol, mg/dL | | | | | | | | |
| HDL cholesterol, mg/dL | 0.006 | 0.002 | - 0.010 | 0.009 | 0.006 | 0.002 | - 0.010 | 0.01 |
| LDL cholesterol, mg/dL | 0.002 | 0.000 | - 0.004 | 0.044 | 0.002 | 0.000 | - 0.004 | 0.025 |
| Triglyceride, mg/dL | 0.001 | 0.001 | - 0.001 | <0.001 | 0.001 | 0.001 | - 0.001 | <0.001 |
| Creatinine, mg/dL | 0.367 | 0.020 | - 0.714 | 0.038 | 0.393 | 0.050 | - 0.736 | 0.025 |
| eGFR, ml/min/1.73 m$^2$ | | | | | | | | |
| hs-CRP, mg/dL | | | | | | | | |
| Fasting glucose, mg/dL | | | | | | | | |
| HbA1C, % | 0.105 | 0.017 | - 0.193 | 0.02 | 0.100 | 0.012 | - 0.186 | 0.027 |
| 10-year ASCVD risk | | | | | | | | |
| Baseline CAC score | | | | | 0.002 | 0.002 | - 0.002 | <0.001 |

HDL: high density lipoprotein; LDL: low density lipoprotein; eGFR: estimated glomerular filtration rate; hs-CRP: high-sensitive C-reactive protein; CAC: coronary artery calcium; ASCVD: atherosclerotic cardiovascular disease; HbA1c, glycated haemoglobin

[a]Model 1 adjusted for male sex, age, waist circumference, hypertension, diabetes, hyperlipidaemia, current smoking, LDL cholesterol, HDL cholesterol, triglyceride, creatinine, and HbA1c.

[b]Model 2 adjusted for the baseline CAC score in addition to the variables in Model 1.

change in CAC progression over time. The probability of CAC progression was non-linear, and the trend gradually diverged as time progressed.

In the present study, men demonstrated a higher proportion of conventional cardiovascular risk factors and higher 10-year ASCVD risk than did women. Therefore, not surprisingly, the baseline CAC score was significantly higher in men than in women. This supports the previous hypothesis that a higher burden of cardiovascular risk factors and more severe baseline coronary atherosclerosis in men contributes to the rapid progression of coronary atherosclerosis [22]. When we analysed sex differences in CAC progression according to clinical risk profiles, we found that the probability of CAC progression at 5 years in men was approximately twice that in women for the lowest 10-year ASCVD risk group. However, the gap between the probability curves was decreased in the intermediate risk group, and the curves finally merged in the highest risk group. Additionally, male sex was a significant predictor for CAC progression only in the lowest risk group. Thus, atherosclerosis progression is similar for both sexes, among those under multiple risk factors, despite an underlying sex difference. Consistent with this, the CONFIRM study reported that after propensity score matching, men and women with no or non-obstructive CAD exhibited the same rates of mortality and myocardial infarction [23]. The present data suggest that men and women at comparably high risk levels

experience coronary atherosclerosis progression in a similar manner, before the onset of adverse cardiac events. Therefore, women with a high ASCVD risk profile should be screened for the presence of CAC and managed with a greater degree of attention [24].

The retrospective observational design of our study introduced several limitations. First, the present study cohort of self-referred healthy individuals may not be fully representative of the general population, and the risk of selection bias must be considered. Even though this was a large multicentre study, the number of enrolled women was smaller than that for men, and there were relatively smaller numbers of women in the higher risk groups, which may have limited the representativeness in the subgroup analyses. This reflects the fact that women seek fewer medical services than do men in this self-referred setting. Second, because of the absence of a specific study protocol guiding follow-up scanning, the interscan duration was relatively short [2.7 years (interquartile range, 1.9–4.1 years)] and was not constant. Nevertheless, the median duration between the initial and last scans did not differ between women and men. Furthermore, to minimize the potential influence of variations in the interscan duration, we analysed the association of annualized CAC progression with various cardiovascular risk factors, including male sex. Additionally, we lacked data regarding menopause and hormone replacement therapy. As postmenopausal women are under atherogenic conditions [25], it would be valuable to compare the CAC progression between postmenopausal women and age-matched men. Finally, since we lacked detailed information regarding medication use, we could not include statin use in the multivariable analysis. Considering the emerging evidence suggesting that statins impact the increase in CAC, further studies are desired to evaluate whether the prominent CAC progression in men than in women is associated with greater statin use.

## Conclusion

The present study demonstrated a sex difference in CAC progression in a large number of asymptomatic individuals. Although the probability of CAC progression was higher in men than in women, subgroup analyses according to the 10-year ASCVD risk demonstrated that this difference diminished as the 10-year ASCVD risk increased.

## Supporting information

**S1 Fig. Kaplan-Meier 5-year CAC progression for men versus women in the propensity matched cohort.** CAC, coronary artery calcium.
(TIF)

**S2 Fig. Kaplan-Meier 5-year CAC progression for men versus women in the propensity matched cohort according to the 10-year ASCVD risk score.** (A) 10-year ASCVD risk <5%, (B) 5%≤ 10-year ASCVD risk <7.5%, (C) 10-year ASCVD risk ≥7.5%. CAC, coronary artery calcium; ASCVD, atherosclerotic cardiovascular disease.
(TIF)

## Author Contributions

**Conceptualization:** Wonjae Lee, Yeonyee E. Yoon, Hyuk-Jae Chang.

**Data curation:** Heesun Lee, Hyo Eun Park, Eun Ju Chun, Hyung-Kwan Kim, Su-Yeon Choi, Sung Hak Park, Hae-Won Han, Jidong Sung, Hae Ok Jung.

**Formal analysis:** Wonjae Lee, Yeonyee E. Yoon, Sang-Young Cho, In-Chang Hwang, Sun-Hwa Kim.

**Investigation:** Su-Yeon Choi.

**Methodology:** Wonjae Lee, In-Chang Hwang, Sun-Hwa Kim, Eun Ju Chun.

**Project administration:** Yeonyee E. Yoon, Hyo Eun Park, Eun Ju Chun, Sung Hak Park, Hae-Won Han, Jidong Sung, Hae Ok Jung.

**Supervision:** Yeonyee E. Yoon, Hyung-Kwan Kim, Goo-Yeong Cho, Hyuk-Jae Chang.

**Validation:** Sang-Young Cho, Sun-Hwa Kim, Heesun Lee.

**Writing – original draft:** Wonjae Lee.

**Writing – review & editing:** Yeonyee E. Yoon, In-Chang Hwang, Goo-Yeong Cho, Hyuk-Jae Chang.

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
