## [Decision Letter · Decision Letter 0]

23 Nov 2020

PONE-D-20-28584

Sex Differences in Coronary Artery Calcium Progression: the Korea Initiatives on Coronary Artery Calcification (KOICA) Registry

PLOS ONE

Dear Dr. Yoon,

Thank you for submitting your manuscript to PLOS ONE. After careful consideration, we feel that it has merit but does not fully meet PLOS ONE’s publication criteria as it currently stands. Therefore, we invite you to submit a revised version of the manuscript that addresses the points raised during the review process.

As you will recognize from the comments of the reviewer major points of critique were raise, especially regarding design of the study and presentation of data.

Please submit your revised manuscript within 2 months. If you will need more time than this to complete your revisions, please reply to this message or contact the journal office at plosone@plos.org. Please include the following items when submitting your revised manuscript:

We look forward to receiving your revised manuscript.

Kind regards,

Rudolf Kirchmair

Academic Editor

PLOS ONE

Journal Requirements:

https://ir.ymlib.yonsei.ac.kr/handle/22282913/170364

The text that needs to be addressed involves the Fig 2 caption, as well as sentences 3-5 of Page 19.

In your revision ensure you cite all your sources (including your own works), and quote or rephrase any duplicated text outside the methods section. Further consideration is dependent on these concerns being addressed.

Reviewers' comments:

Reviewer's Responses to Questions

**Comments to the Author**

1. Is the manuscript technically sound, and do the data support the conclusions?

Reviewer #1: Yes

2. Has the statistical analysis been performed appropriately and rigorously? 

Reviewer #1: Yes

3. Have the authors made all data underlying the findings in their manuscript fully available?

Reviewer #1: No

4. Is the manuscript presented in an intelligible fashion and written in standard English?

Reviewer #1: Yes

5. Review Comments to the Author

Reviewer #1: This is an interesting study examining sex differences in CAC progression. The study has a number of strengths including large sample size of 9,675 men and 1,709 women with follow-up CAC scores. The authors did multivariate analysis as well as propensity matching to examine CAC progression rate in men vs women.

An important finding is that among the high-risk groups by ASCVD score, that men and women had similar rates of CAC progression, and the male predominance of progression was only more pronounced in the lower risk groups

Overall I liked this paper and think the findings are important in adding to our understanding of CAC progression among men and women

A few comments.

1. Abstract – please state how progression of CAC is defined. As progression is likely more to occur among those with baseline disease, please also state prevalence of baseline CAC >0 among men and women, or median CAC score in abstract. Men might be more likely to progress is they have more disease to start with at baseline.

2. Did you exclude people with known clinical ASCVD? Was this a population of asymptomatic individuals? I would assume so, as the 10-year risk score applies in primary prevention not secondary prevention, but it does not explicitly state that this was a population without known clinical ASCVD.

3. Time is a big risk factor for progression, need to account for time between CT scans. Individuals who had the 2 CTs close together will be less likely to have progression than if CTs were farther apart. The authors used annualized difference which I think is appropriate as it adjusts for time.

4. I am also glad the adjusted for baseline CAC, because presence or absence of baseline CAC is a driver of CAC progression.

5. Can the authors add use of statins to Table 1? What was the use of statins in this population. Statin use has been shown to actually increase the CAC score despite its known risk reduction in CVD events (statins likely transform softer plaques into more stable dense plaques). The models should adjust for statin use.

6. Table 3 – change Male gender to Male sex. Sex is the more appropriate term here than gender since you are referring to likely biological differences related to sex hormones and other biological factors.

7. Also for table 3, include a footnote about what Model 1 and Model 1 adjusted for.

8. It is disappointing that there was no menopause data in this cohort, but this was appropriately acknowledged as limitation by the authors.

9. A major limitation of the study design is that this is self-referral cohort not a population based study, so there is referral bias – but this was acknowledged by authors. Men might be more likely to be referred for a second CAC scan compared to men, and indeed

10. Another limitation that should be mentioned is the relatively short followup time between scans, and sex differences in CAC progression over a longer period (i.e >10 years) could not be examined but would be of interest.

11. This is likely beyond the scope of this paper- but perhaps for the next paper, I am interested in knowing whether CAC progression is associated with incident ASCVD events incremental to risk conferred by baseline CAC, and if so, whether that association differed by sex. Some studies but not all have shown that an elevated CAC score in women confers greater CVD risk than it does in men. So is CAC progression in women also associated with greater CVD risk than in men?

6. PLOS authors have the option to publish the peer review history of their article (what does this mean?). If published, this will include your full peer review and any attached files.

Reviewer #1: No

---

## [Author Response · Author response to Decision Letter 0]

4 Jan 2021

Review Comments to the Author

Reviewer #1

This is an interesting study examining sex differences in CAC progression. The study has a number of strengths including large sample size of 9,675 men and 1,709 women with follow-up CAC scores. The authors did multivariate analysis as well as propensity matching to examine CAC progression rate in men vs women. An important finding is that among the high-risk groups by ASCVD score, that men and women had similar rates of CAC progression, and the male predominance of progression was only more pronounced in the lower risk groups. Overall I liked this paper and think the findings are important in adding to our understanding of CAC progression among men and women

1. Abstract – please state how progression of CAC is defined. As progression is likely more to occur among those with baseline disease, please also state prevalence of baseline CAC >0 among men and women, or median CAC score in abstract. Men might be more likely to progress is they have more disease to start with at baseline.

Re: Thank you for your valuable suggestion. As recommended, we have added the definition of CAC progression and the prevalence of baseline CAC>0 at baseline to the abstract as follows:

Page 3, lines 53-56: At baseline, men were more likely to have a CAC score >0 than were women (47.8% vs. 28.6%). The probability of CAC progression at 5 years, defined as [√CAC score (follow-up) - √CAC score (baseline)] ≥2.5, was 47.4% in men and 29.7% in women (p<0.001). 

2. Did you exclude people with known clinical ASCVD? Was this a population of asymptomatic individuals? I would assume so, as the 10-year risk score applies in primary prevention not secondary prevention, but it does not explicitly state that this was a population without known clinical ASCVD.

Re: Thank you for your valuable comment. First, we would like to explain how Korean health check-up centres work. Once every two years, Koreans may undergo health check-ups that are fully covered by the national health insurance system; however, if they wish, they can undergo the health check-up at specialized health check-up centres that do not receive any insurance coverage. In the former case, CAC scoring is not included in the health check-up program, whereas, in the latter case, individuals can choose the tests performed, including CAC scoring, and the costs are borne by the patients or their employer. Of course, patients with prior ASCVD can visit health check-up centres in a self-referral setting. However, most health check-up centres do not recommend CAC scoring in patients with known or suspected ASCVD, referring such patients to specialists. Therefore, the KOICA registry, with participants recruited from six healthcare centres, comprises apparently healthy individuals who underwent CAC scanning as a part of a health check-up.

 When we evaluated prior ASCVD based on the responses to self-administered questionnaires, 683 (6.0%) participants indicated that they were diagnosed with ASCVD. However, considering the real clinical situation in Korea, these self-reported ASCVDs usually do not refer to the presence of angiographically proven significant CAD or ischemic stroke diagnosed based on brain MRI, but rather refer to the experience of atypical chest pain, facial palsy, or numbness, etc. Furthermore, considering that most health check-up centres do not recommended CAC scoring for patients with clinically significant ASCVD, we did not exclude patients with self-reported ASCVD. Nevertheless, when we performed the primary statistical analyses after the exclusion of participants with self-reported ASCVD (n=683) and those who did not indicate whether or not they had prior ASCVD (n=292), the results were not significantly changed. The figures corresponding to Fig 3 and Fig 4-C in the manuscript file, as well as the table corresponding to Table 3, are shown below. Therefore, we consider our decision to include these individuals to be reasonable.

 

Response Letter Figure 1 (corresponding to Fig 3 in the manuscript). Kaplan-Meier 5-year CAC progression for men versus women in asymptomatic Korean individuals -> refer to the attached file

Response Letter Figure 2 (corresponding to Fig 4A-C in the manuscript). Kaplan-Meier 5-year CAC progression for men versus women according to the 10-year ASCVD risk score. (A) 10-year ASCVD risk <5%, (B) 5%≤ 10-year ASCVD risk <7.5%, (C) 10-year ASCVD risk ≥7.5%. -> refer to the attached file

Response Letter Table 1 (corresponding to Table 3 in the manuscript). Multivariate analysis for factors associated with the CAC progression rate -> refer to the attached file

HDL: high density lipoprotein; LDL: low density lipoprotein; eGFR: estimated glomerular filtration rate; hs-CRP: high-sensitive C-reactive protein; CAC: coronary artery calcium; ASCVD: atherosclerotic cardiovascular disease; HbA1c, glycated haemoglobin

aModel 1 adjusted for male sex, age, waist circumference, hypertension, diabetes, hyperlipidaemia, current smoking, LDL cholesterol, HDL cholesterol, triglyceride, creatinine, and HbA1c.

bModel 2 adjusted for the baseline CAC score in addition to the variables in Model 1.

3. Time is a big risk factor for progression, need to account for time between CT scans. Individuals who had the 2 CTs close together will be less likely to have progression than if CTs were farther apart. The authors used annualized difference which I think is appropriate as it adjusts for time.

Re: We share the reviewer’s concern regarding the potential effect of variations in the interscan duration. Although the interscan duration in men [median 2.8 years, (IQR 1.9-4.1)] was similar to that in women [median 2.6 years (IQR 1.9-4.1)], we took great care to minimize the effect of variations in the interscan duration. Specifically, we calculated the annualized CAC progression rate and evaluated the association between the annualized CAC progression rate and various cardiovascular risk factors, including male sex. This information has been added to the limitations section.

Page 21, lines 314-321: Furthermore, to minimize the potential influence of variations in the interscan duration, we analysed the association of annualized CAC progression with various cardiovascular risk factors, including male sex.

4. I am also glad the adjusted for baseline CAC, because presence or absence of baseline CAC is a driver of CAC progression.

Re: Thank you for pointing out this oversight. As you have pointed out, previous studies demonstrated that baseline CAC independently predicts CAC progression. (Atherosclerosis 2014;232:339–345). In the current study, there was a difference in baseline CAC score between men and women. Furthermore, baseline CAC was independently associated with the annualized CAC progression rate, as expected. Considering the effect of the baseline CAC score, we used two different multivariate models, with and without the baseline CAC score, to evaluate whether the association between male sex and CAC progression was affected by the baseline CAC score (Table 3). We have added footnotes to Table 3 to clarify the two models (see the response to #7).

5. Can the authors add use of statins to Table 1? What was the use of statins in this population. Statin use has been shown to actually increase the CAC score despite its known risk reduction in CVD events (statins likely transform softer plaques into more stable dense plaques). The models should adjust for statin use.

Re: We greatly appreciate your comment. We completely agree that it is very important to consider statin’s effect on CAC progression. However, unfortunately, as the questionnaire was self-administered, the collected medication history was incomplete; only 20% of study participants answered the question regarding statin use. Considering the low response rate reading medication usage, we did not include statin use in the analysis in the current study. Nevertheless, as this is an important issue, we have added the following sentence to the limitations section:

Page 21, lines 318-322: Finally, since we lacked detailed information regarding medication use, we could not include statin use in the multivariable analysis. Considering the emerging evidence suggesting that statins impact the increase in CAC, further studies are desired to evaluate whether the prominent CAC progression in men than in women is associated with greater statin use. 

6. Table 3 – change Male gender to Male sex. Sex is the more appropriate term here than gender since you are referring to likely biological differences related to sex hormones and other biological factors.

Re: Thank you for your valuable comment. We agree with the reviewer and prefer “male sex” over “male gender”. We have modified Table 3 accordingly (see the response to comment #7). 

7. Also for table 3, include a footnote about what Model 1 and Model 1 adjusted for.

Re: Thank you for your suggestion. We have added footnotes as follows:

Table 3. Multivariate analysis for factors associated with the CAC progression rate -> refer to the attached file

HDL: high density lipoprotein; LDL: low density lipoprotein; eGFR: estimated glomerular filtration rate; hs-CRP: high-sensitive C-reactive protein; CAC: coronary artery calcium; ASCVD: atherosclerotic cardiovascular disease; HbA1c, glycated haemoglobin

aModel 1 adjusted for male sex, age, waist circumference, hypertension, diabetes, hyperlipidaemia, current smoking, LDL cholesterol, HDL cholesterol, triglyceride, creatinine, and HbA1c.

bModel 2 adjusted for the baseline CAC score in addition to the variables in Model 1.

　 

10. Another limitation that should be mentioned is the relatively short follow-up time between scans, and sex differences in CAC progression over a longer period (i.e >10 years) could not be examined but would be of interest.

Re: Thank you for your valuable comment. We agree that examining the long-term trend in CAC progression, as well as sex differences in this trend, would be a very interesting topic. Therefore, we have added the following sentence to the limitations section:

Page 21, lines 308-314: Second, because of the absence of a specific study protocol guiding follow-up scanning, the interscan duration was relatively short [2.7 years (interquartile range, 1.9-4.1 years)] and was not constant. Nevertheless, the median duration between the initial and last scans did not differ between women and men. Furthermore, to minimize the potential influence of variations in the interscan duration, we analysed the association of annualized CAC progression with various cardiovascular risk factors, including male sex. 

11. This is likely beyond the scope of this paper- but perhaps for the next paper, I am interested in knowing whether CAC progression is associated with incident ASCVD events incremental to risk conferred by baseline CAC, and if so, whether that association differed by sex. Some studies but not all have shown that an elevated CAC score in women confers greater CVD risk than it does in men. So is CAC progression in women also associated with greater CVD risk than in men?

Re: We appreciated the opportunity to discuss this interesting topic with you. To our knowledge, CAC progression is known to be associated with incident ASCVD events, even after adjustment for baseline CAC (JACC Cardiovascular Imaging 2016;9(12):1420-1429). However, as you mentioned, whether the association differs by sex was not evaluated. It would be noteworthy to investigate whether CAC progression in women has a similar impact on the risk of ASCVD to that in men. In fact, KOICA investigators are currently preparing an outcome study; thus, we hope that we will be able to answer your question in our next project.

---

## [Decision Letter · Decision Letter 1]

8 Mar 2021

Sex Differences in Coronary Artery Calcium Progression: the Korea Initiatives on Coronary Artery Calcification (KOICA) Registry

PONE-D-20-28584R1

Dear Dr. Yoon,

We’re pleased to inform you that your manuscript has been judged scientifically suitable for publication and will be formally accepted for publication once it meets all outstanding technical requirements.

Kind regards,

Rudolf Kirchmair

Academic Editor

PLOS ONE

Additional Editor Comments (optional):

Reviewers' comments:

Reviewer's Responses to Questions

**Comments to the Author**

1. If the authors have adequately addressed your comments raised in a previous round of review and you feel that this manuscript is now acceptable for publication, you may indicate that here to bypass the “Comments to the Author” section, enter your conflict of interest statement in the “Confidential to Editor” section, and submit your "Accept" recommendation.

Reviewer #1: All comments have been addressed

2. Is the manuscript technically sound, and do the data support the conclusions?

Reviewer #1: (No Response)

3. Has the statistical analysis been performed appropriately and rigorously? 

Reviewer #1: Yes

4. Have the authors made all data underlying the findings in their manuscript fully available?

Reviewer #1: No

5. Is the manuscript presented in an intelligible fashion and written in standard English?

Reviewer #1: Yes

6. Review Comments to the Author

Reviewer #1: All of my prior comments have been sufficiently addressed by authors. Thank you. No further comments from me at this time.

The authors have indicated that data would be made available upon request after review by IRB and with data sharing agreement, and I agree with that. Many other cohorts (such as MESA) have similar restrictions about sharing data. I do not see a problem with this approach.

7. PLOS authors have the option to publish the peer review history of their article (what does this mean?). If published, this will include your full peer review and any attached files.

Reviewer #1: No

---

## [Editor Report · Acceptance letter]

31 Mar 2021

PONE-D-20-28584R1 

Sex differences in coronary artery calcium progression: the Korea Initiatives on Coronary Artery Calcification (KOICA) registry 

Dear Dr. Yoon:

I'm pleased to inform you that your manuscript has been deemed suitable for publication in PLOS ONE. Congratulations! Your manuscript is now with our production department. 

Kind regards, 

on behalf of

Prof Rudolf Kirchmair 

Academic Editor

PLOS ONE